# GT-GAN: General Purpose Time Series Synthesis with Generative Adversarial Networks

**Jinsung Jeon**
Yonsei University
jjsjjs0902@yonsei.ac.kr

**Jeonghak Kim**
Kakao Corp.
haggie.pro@kakaocorp.com

**Haryong Song**
Linger Studio Corp.
harong@lingercorp.com

**Seunghyeon Cho**
Yonsei University
seunghyeoncho@yonsei.ac.kr

**Noseong Park**
Yonsei University
noseong@yonsei.ac.kr

## Abstract

Time series synthesis is an important research topic in the field of deep learning, which can be used for data augmentation. Time series data types can be broadly classified into regular or irregular. However, there are no existing generative models that show good performance for both types without any model changes. Therefore, we present a general purpose model capable of synthesizing regular and irregular time series data. To our knowledge, we are the first designing a general purpose time series synthesis model, which is one of the most challenging settings for time series synthesis. To this end, we design a generative adversarial network-based method, where many related techniques are carefully integrated into a single framework, ranging from neural ordinary/controlled differential equations to continuous time-flow processes. Our method outperforms all existing methods.

## 1 Introduction

Time series data occurs frequently in real-world applications [Reinsel, 2003, Fu, 2011, Li et al., 2018, Yu et al., 2018, Wu et al., 2019, Guo et al., 2019, Bai et al., 2019, Song et al., 2020, Huang et al., 2020a, Ren et al., 2021, Tekin et al., 2021]. Among many tasks related to time series, synthesizing time series data is one of the most important tasks because real-world time series data is frequently imbalanced and/or insufficient. Since regular and irregular time series data have different characteristics, however, different model designs had been adopted for them. Therefore, existing time series synthesis work focuses on either regular or irregular time series synthesis [Yoon et al., 2019, Alaa et al., 2021]. To our knowledge, there are no existing methods that work well for both types.

Regular time series means regularly sampled observations without any missing ones, and irregular times series means that some observations are missing from time to time. Irregular time series is much harder to process than regular time series. For instance, it is known that neural networks perform better after transforming time series data into its frequency domain, i.e., the Fourier transform, and some time series generative models use this approach [Alaa et al., 2021]. However, it is not easy to observe pre-determined frequencies from highly irregular time series [Kidger et al., 2019]. However, continuous-time models [Chen et al., 2018, Kidger et al., 2020, Brouwer et al., 2019] show good performance in processing both regular and irregular time series. By resorting to them, we propose a general purpose model that can synthesize both time series types without any model changes.

To achieve the goal, we design a sophisticated model which utilizes a diverse set of technologies, ranging from generative adversarial networks (GANs [Goodfellow et al., 2014]), and autoencoders (AEs) to neural ordinary differential equations (NODEs [Chen et al., 2018]), neural controlled differ-

36th Conference on Neural Information Processing Systems (NeurIPS 2022).

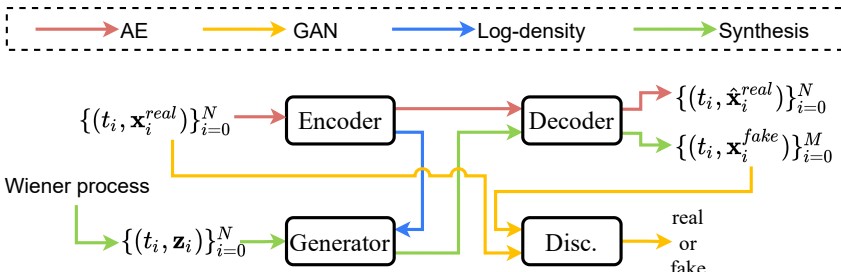

Figure 1: The overall design of our method, GT-GAN. Each color means a separate workflow. The autoencoder and the GAN share the workload to synthesize regular and irregular time series.

ential equations (NCDEs [Kidger et al., 2020]), and continuous time-flow processes (CTFPs [Deng et al., 2020]), which reflects the difficulty of the problem.

Fig. 1 shows the overall design of our proposed method. One of the key points in our model design is that we combine the *adversarial* training of GANs and the *exact maximum likelihood* training of CTFPs into a single framework. However, the exact maximum likelihood training is applicable only to invertible mapping functions whose input and output sizes are the same. Therefore, we design an *invertible* generator, and adopt an autoencoder, on whose hidden space our GAN performs the adversarial training.

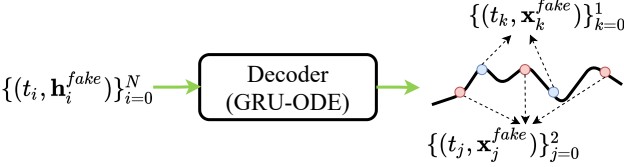

Figure 2: An example of sampling two irregular time series from a fake continuous path represented by an ordinary differential equation (ODE). In other words, we solve ODE problems to sample regular/irregular time series.

In other words, i) the hidden vector size of the encoder is the same as the noisy vector of the generator, ii) the generator produces a set of fake hidden vectors, iii) the decoder converts the set into a fake continuous path (cf. Fig. 2), and iv) the discriminator provides feedback after reading the sampled fake sample. We note that in the third step, a fake continuous path is created by the decoder. Therefore, we can sample any arbitrary regular/irregular time series sample from the fake path, which shows the flexibility in our method.

We conduct experiments with 4 datasets and 7 baselines. Since our method is able to support both the regular and irregular time series synthesis, we test for both of them. Our method outperforms other baselines in both environments. Our contributions can be summarized as follows:

1. We design a model based on various state-of-the-art deep learning technologies. Our method is able to process any types of time series data, ranging from regular to irregular, without any model changes.

2. Our experimental results and visualization prove the efficacy of the proposed model.

3. Since our task is one of the most challenging tasks for time series synthesis, the proposed model architecture is carefully designed. Our ablation studies show that our proposed model does not work well if any part is missing.

## 2 Related work and preliminaries

GANs are one of the most representative generative technology. Ever since the first introduction in its seminal research paper, GANs have been adopted to main different domains. Recently, researchers focused on their synthesis for time series data. Therefore, there have been proposed several GANs for time series synthesis. C-RNN-GAN [Mogren, 2016] has a regular GAN framework that can be applied to sequential data by using LSTM in its generator and discriminator. Recurrent Conditional GAN (RCGAN [Esteban et al., 2017]) took a similar approach except that its generator and discriminator take conditional input for better synthesis. WaveNet [van den Oord et al., 2016] also generates time series data from the conditional probability of previous data by using the dilated casual convolution.

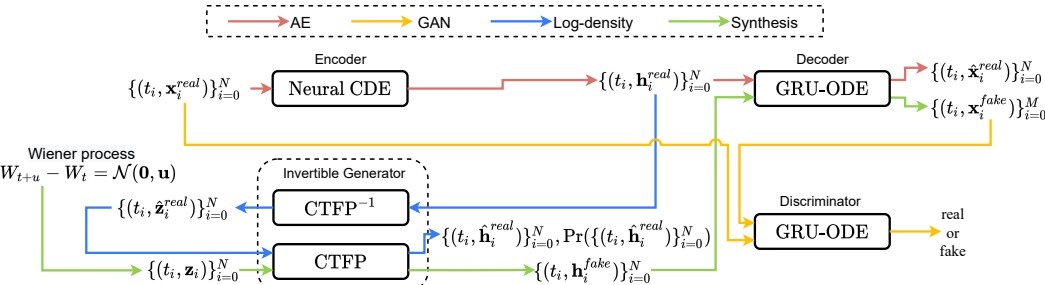

Figure 3: The detailed architecture of our proposed method. Neural CDEs (or NCDEs) are a recent breakthrough for processing time series. GRU-ODEs are a continuous interpretation of gated recurrent units (GRUs) based on NODEs. CTFPs are a flow-based concept to convert an input time series process into a target process. CTFPs are not a GAN-based concept but we integrate them into our framework, considering the challenging nature of the general purpose time series synthesis.

WaveGAN [Donahue et al., 2019] has a similar approach with DCGAN [Radford et al., 2016], where its generator is based on WaveNet. We can modify the teacher-forcing (T-Forcing [Graves, 2014]) and professor-forcing (P-Forcing [Lamb et al., 2016]) models to generate time series data from noise vectors, although they are not GAN models, by using the forecasting characteristic of those models.

TimeGAN [Yoon et al., 2019] is yet another model for time series synthesis. This model aims mainly at synthesizing fake *regular* time series samples. They proposed a framework where the adversarial training of GANs and the supervised training of predicting $\mathbf{x}_{i+1}$ from $\mathbf{x}_i$, where $\mathbf{x}_i$ and $\mathbf{x}_{i+1}$ mean two multivariate time series values at time $t_i$ and $t_{i+1}$, respectively.

## 3 Proposed method

In this section, we describe our design. Since our general purpose time series synthesis is a challenging task, the proposed design is much more complicated than other baselines.

### 3.1 Overall workflow

We first describe the overall workflow in our model design, which consists of several different data paths (and several different training methods based on the data paths) as follows:

1. **Autoencoder path:** Given an time series sample $\{(t_i, \mathbf{x}_i^{real})\}_{i=0}^N$, the encoder produces a set of hidden vectors $\{\mathbf{h}_i^{real}\}_{i=0}^N$. The decoder recovers a continuous path $\hat{X}^{real}$, which enhances the flexibility of our proposed method. From the path $\hat{X}^{real}$, we sample $\{(t_i, \hat{\mathbf{x}}_i^{real})\}_{i=0}^N$. We train the encoder and the decoder using the standard autoencoder (AE) loss to match $\mathbf{x}_i^{real}$ and $\hat{\mathbf{x}}_i^{real}$ for all $i$.

2. **Adversarial path:** Given a set of noisy vectors $\{\mathbf{z}_i\}_{i=0}^N$, our generator produces a set of fake hidden vectors $\{\mathbf{h}_i^{fake}\}_{i=0}^N$. The decoder recovers a fake continuous path $\hat{X}^{fake}$ from $\{\mathbf{h}_i^{fake}\}_{i=0}^N$. We sample $\{(t_j, \mathbf{x}_j^{fake})\}_{j=0}^M$ from $\hat{X}^{fake}$ and feed it into the discriminator. For irregular time series synthesis, we sample $t_j$ in $[0, T]$. We train the generator, the decoder, and the discriminator using the standard adversarial loss.

3. **Log-density path:** Given a set of hidden vectors $\{\mathbf{h}_i^{real}\}_{i=0}^N$ for an time series sample $\{(t_i, \mathbf{x}_i^{real})\}_{i=0}^N$, the inverse path of the generator reproduces a set of noisy vectors $\{\hat{\mathbf{z}}_i\}_{i=0}^N$. We feed $\{\hat{\mathbf{z}}_i\}_{i=0}^N$ into its forward path again to reproduce $\{\hat{\mathbf{h}}_i^{real}\}_{i=0}^N$, where $\hat{\mathbf{h}}_i^{real} = \mathbf{h}_i^{real}$ for all $i$. During the forward pass, we calculate the negative log probability of $-\log p(\hat{\mathbf{h}}_i^{real})$ for all $i$ with the change of variable theorem and minimize it for training, being inspired by Grover et al. [2018] and Deng et al. [2020].

In particular, we note that the dimensionality of the hidden space in the autoencoder is the same as that of the latent input space of the generator, i.e., $\dim(\mathbf{h}) = \dim(\mathbf{z})$. This is needed for the exact likelihood training in the generator — the change of variable theorem requires that the input and

output sizes are the same to estimate the exact likelihood. In addition to it, we let the autoencoder and the generator share the workload to synthesize fake time series by combining them into a single framework, i.e., the generator synthesizes fake hidden vectors and the decoder reproduces human-readable fake time series from them.

## 3.2 Autoencoder

**Encoder** General NCDEs, which are considered as a continuous analogue to recurrent neural networks (RNNs), are defined as follows:

$$
\begin{aligned}
\mathbf{h}(t_{i+1}) &= \mathbf{h}(t_i) + \int_{t_i}^{t_{i+1}} f(\mathbf{h}(t); \theta_f) dX(t) \\
&= \mathbf{h}(t_i) + \int_{t_i}^{t_{i+1}} f(\mathbf{h}(t); \theta_f) \frac{dX(t)}{dt} dt,
\end{aligned}
\tag{1}
$$

where $X(t)$ is a continuous path created from a raw discrete time series sample $\{(t_i, \mathbf{x}_i^{real})\}_{i=0}^N$ by an interpolation algorithm — we note that $X(t_i) = (t_i, \mathbf{x}_i^{real})$ for all $i$, and for other non-observed time-points the interpolation algorithm fills out values. Note that NCDEs keep reading the time-derivative of $X(t)$, denoted $\dot{X}(t) \stackrel{\text{def}}{=} \frac{dX(t)}{dt}$. In our case, we collect $\{\mathbf{h}_i^{real}\}_{i=0}^N$ as follows:

$$
\mathbf{h}_{i+1}^{real} = \mathbf{h}_i^{real} + \int_{t_i}^{t_{i+1}} f(\mathbf{h}(t); \theta_f) \frac{dX(t)}{dt} dt,
\tag{2}
$$

where $\mathbf{h}_0^{real} = \texttt{FC}_{\dim(\mathbf{x}) \to \dim(\mathbf{h})}(\mathbf{x}_0^{real})$ and $\texttt{FC}_{input\_size \to output\_size}$ is a fully-connected layer with specific input and output sizes. We refer to Appendix B for the ODE function $f$ definition.

Therefore, the input time series $\{(t_i, \mathbf{x}_i^{real})\}_{i=0}^N$ is represented by a set of hidden vectors $\{(t_i, \mathbf{h}_i^{real})\}_{i=0}^N$. Because NCDEs are a continuous analogue to RNNs, it shows the best fit to processing irregular time series [Kidger et al., 2020].

**Decoder** Our decoder, which reproduces a time series from its hidden representations, is based on GRU-ODEs [Brouwer et al., 2019] and is defined as follows:

$$
\bar{\mathbf{d}}(t_{i+1}) = \mathbf{d}(t_i) + \int_{t_i}^{t_{i+1}} g(\mathbf{d}(t), t; \theta_g) dt,
\tag{3}
$$

$$
\mathbf{d}(t_{i+1}) = \texttt{GRU}(\mathbf{h}_{i+1}, \bar{\mathbf{d}}(t_{i+1})),
\tag{4}
$$

$$
(t_{i+1}, \hat{\mathbf{x}}_{i+1}) = (t_{i+1}, \texttt{FC}_{\dim(\mathbf{d}) \to \dim(\mathbf{x})}(\mathbf{d}(t_{i+1}))),
\tag{5}
$$

where $\mathbf{d}(t_0) = \texttt{FC}_{\dim(\mathbf{h}) \to \dim(\mathbf{d})}(\mathbf{h}_0)$ and $\mathbf{h}_i$ means either the $i$-th real or fake hidden vector, i.e., $\mathbf{h}_i^{real}$ or $\mathbf{h}_i^{fake}$ — recall that in Fig. 3, the decoder is involved in both the autoencoder and the synthesis processes. $\hat{\mathbf{x}}$ means a reproduced copy of $\mathbf{x}$. GRU-ODEs uses the technology called neural ordinary differential equations (NODEs) to *continuously* interpret GRUs and we refer to Appendix B for the ODE function $g$ definition.

In particular, the gated recurrent unit (GRU) at Eq. (4) is called as *jump* which is known to be effective in processing time series with NODEs [Brouwer et al., 2019, Jia and Benson, 2019]. We train the encoder-decoder using the standard reconstruction loss between $\mathbf{x}_i^{real}$ and $\hat{\mathbf{x}}_i^{real}$ for all $i$ in all training time series samples.

## 3.3 Generative adversarial network

**Generator** Whereas generators typically read a noisy vector to generate a fake sample in standard GANs, our generator reads a continuous path (or time series) sampled from a Wiener process to generate a fake time series sample — this generation concept is known as continuous time flow processes (CTFPs [Deng et al., 2020]). Appendix. B.3 shows an example of our generation process. The input to our generation process is a random path sampled from a Wiener process, which is represented by a time series of latent vectors $\{(t_i, \mathbf{z}_i)\}_{i=0}^N$ in the path, and the output is a path of hidden vectors which is also represented by a time series of hidden vectors $\{(t_i, \mathbf{h}_i^{fake})\}_{i=0}^N$.

Therefore, our generator can be written as follows:

$$\mathbf{h}_i^{fake} = \mathbf{w}_i(1) = \mathbf{w}_i(0) + \int_0^1 r(\mathbf{w}_i(\tau), a_i(t), t; \theta_r)d\tau, \tag{6}$$

where $\mathbf{w}_i(0) = \mathbf{z}_i$, $a_i(0) = t_i$. Here, $\tau$ means a virtual time variable of the integral problem, and $t_i$ is a real physical time contained in a time series sample $\{(t_i, \mathbf{x}_i^{real})\}_{i=0}^M$. We note that this design corresponds to a NODE model augmented with $a_i(t)$. We refer to Appendix B for the ODE function $r$ definition.

Owing to the invertible nature of NODEs, we can calculate the exact log-density of $\mathbf{h}_i^{real}$, i.e., the probability that $\mathbf{h}_i^{real}$ is generated by the generator, using the change of variable theorem and the Hutchinson's stochastic trace estimator as follows [Grathwohl et al., 2019, Deng et al., 2020]:

$$\hat{\mathbf{w}}(0) = \mathbf{h}_i^{real} + \int_1^0 r(\mathbf{w}(\tau), a_i(\tau), t; \theta_r)d\tau, \tag{7}$$

$$\log \Pr(\hat{\mathbf{h}}_i^{real}) = \log \Pr(\hat{\mathbf{w}}(0))$$
$$+ \int_0^1 tr\Big(\frac{\partial r(\mathbf{w}(\tau), a_i(\tau), t; \theta_r)}{\partial \mathbf{w}(\tau)}\Big)d\tau, \tag{8}$$

where $\hat{\mathbf{w}}(0)$ means $\hat{\mathbf{z}}_i^{real}$ in Fig. 3. $\hat{\mathbf{h}}_i^{real}$ means a reproduced copy of $\mathbf{h}_i^{real}$ by our generator. Eq. (7) corresponds to "CTFP$^{-1}$", and Eqs. (6) and (8) to "CTFP" in Fig. 3. We note that in Eq. (7), the integral time is reversed to solve the reverse-mode integral problem.

Therefore, we minimize the negative log-density, denoted $-\log \Pr(\hat{\mathbf{h}}_i^{real})$, for each $t_i$, and our generator is trained by the two different training paradigms: i) the adversarial training against the discriminator, and ii) the maximum likelihood estimator (MLE) training with the log-density.

**Discriminator** We design our discriminator based on the GRU-ODE technology as follows:

$$\bar{\mathbf{c}}(t_{i+1}) = \mathbf{c}(t_i) + \int_{t_i}^{t_{i+1}} q(\mathbf{c}(t), t; \theta_q)dt, \tag{9}$$

$$\mathbf{c}(t_{i+1}) = \texttt{GRU}(\mathbf{x}_{i+1}, \bar{\mathbf{c}}(t_{i+1})), \tag{10}$$

where $\mathbf{c}(t_0) = \texttt{FC}_{\dim(\mathbf{x})\to\dim(\mathbf{c})}(\mathbf{x}_0)$, and $\mathbf{x}_i$ means the $i$-th time series value, i.e., $\mathbf{x}_i^{real}$ or $\mathbf{x}_i^{fake}$. The ODE function $q$ has the same architecture as $g$ but with its own parameters $\theta_q$. After that, we calculate the real or fake classification $\mathbf{y} = \sigma(\texttt{FC}_{\dim(\mathbf{c})\to 2}(\mathbf{c}(t_N)))$, where $\sigma$ is a softmax activation.

The role of each part of our proposed model is in Appendix M.

### 3.4 Training method

We use the mean squared reconstruction loss, i.e., the mean of $\|\mathbf{x}_i^{real} - \hat{\mathbf{x}}_i^{real}\|_2^2$ for all $i$, to train the encoder-decoder architecture. Then, we use the standard GAN loss to train the generator and the discriminator. In our preliminary experiments, we found that the original GAN loss is suitable for our task. Instead of other variations, such as WGAN-GP [Gulrajani et al., 2017], therefore, we use the standard GAN loss. We train our model in the following sequence:

1. We pre-train the encoder-decoder networks the reconstruction loss for $K_{AE}$ iterations.
2. After the above pre-training step, we start to jointly train all networks in the following sequence for $K_{JOINT}$ iterations: i) training the encoder-decoder networks with the reconstruction loss, ii) training the discriminator-generator networks with the GAN loss, iii) training the decoder to improve the discriminator's classification output with the discriminator loss and iv) the generator with the MLE loss every $P_{MLE}$ iteration. We found that too frequent MLE training incurs mode-collapse so we use it every $P_{MLE}$ iteration.

In particular, the 2-ii step to train the decoder to help the discriminator out is one additional point where the autoencoder and the GAN are integrated into a single framework. In other words, the generator should deceive both the decoder and the discriminator. Our training algorithm refer to Appendix 1

The well-posedness[1] of NCDEs and GRU-ODEs was already proved in Lyons et al. [2007, Theorem 1.3] and Brouwer et al. [2019] under the mild condition of the Lipschitz continuity. We show that our NCDE layers are also well-posed problems. Almost all activations, such as ReLU, Leaky ReLU, SoftPlus, Tanh, Sigmoid, ArcTan, and Softsign, have a Lipschitz constant of 1. Other common neural network layers, such as dropout, batch normalization and other pooling methods, have explicit Lipschitz constant values. Therefore, the Lipschitz continuity of ODE/CDE functions can be fulfilled in our case. In other words, it is a well-posed training problem. As a result, our training algorithm solves a well-posed problem so its training process is stable in practice.

## 4    Experimental evaluations

Our software and hardware environments are as follows: UBUNTU 18.04 LTS, PYTHON 3.8.10, PYTORCH 1.8.1, TENSORFLOW 2.5.0, CUDA 11.2, and NVIDIA Driver 417.22, i9 CPU, and NVIDIA RTX 3090. The mean and variance of 10 runs are reported for model evaluation.

### 4.1    Experimental environments

**Datasets**    We conduct experiments with 2 simulated and 2 real-world datasets. Sines has 5 features where each feature is created with different frequencies and phases independently. For each feature, $i \in \{1, ..., 5\}$, $x_i(t) = sin(2\pi f_i t + \theta_i)$, where $f_i \sim \mathcal{U}[0, 1]$ and $\theta_i \sim \mathcal{U}[-\pi, \pi]$. MuJoCo is multivariate physics simulation time series data with 14 features. Stocks is the Google stock price data from 2004 to 2019. Each observation represents one day and has 6 features. Energy is a UCI appliance energy prediction dataset with 28 values. To create the challenging irregular environments, 30, 50, 70% of observations from each time series sample $\{(t_i, \mathbf{x}_i^{real})\}_{i=0}^{N}$ is randomly dropped — in other words, $N$ decreases to $0.7N, 0.5N, 0.3N$. Dropping random values has been mainly used to create irregular time series environments in the literature [Kidger et al., 2019, Xu and Xie, 2020, Huang et al., 2020b, Tang et al., 2020, Zhang et al., 2021, Jhin et al., 2021, Deng et al., 2021]. Therefore, we conduct experiments with both the regular and the irregular environments.

**Baselines**    We consider the following baselines for the regular time series experiments: TimeGAN, RCGAN, C-RNN-GAN, WaveGAN, WaveNet, T-Forcing, and P-Forcing. For the irregular experiments, we exclude WaveGAN and WaveNet, which cannot handle irregular time series, and redesign other baselines by replacing their GRU with GRU-$\triangle t$ and GRU-Decay (GRU-D) [Che et al., 2018]. GRU-$\triangle t$ and GRU-D are effective models for processing irregular time series data. GRU-$\triangle t$ additionally uses the time difference between observations as input. GRU-D is a modification of GRU-$\triangle t$ to learnt exponential decays between observations. TimeGAN-$\triangle t$, RCGAN-$\triangle t$, C-RNN-GAN-$\triangle t$, T-Forcing-$\triangle t$, and P-Forcing-$\triangle t$ (resp. TimeGAN-D, RCGAN-D, C-RNN-GAN-D, T-Forcing-D, and P-Forcing-Decay) are modified with GRU-$\triangle t$ (resp. GRU-D) and can handle irregular data.

Our ablation studies also involve many advanced methods, based on NODEs, VAEs, flow models, and so forth. We intentionally leave these advanced methods for our ablation studies since our proposed method internally has them as sub-parts.

**Evaluation metrics**    For quantitative evaluation of synthesized data, it is evaluated with the discriminative score and the predictive score used in TimeGAN [Yoon et al., 2019]. The discriminative score measures the similarity between the original data and the synthesized data. After learning a model that classifies the original data and the synthesized data using a neural network, it is tested whether the original data and the synthesized data are classified well. The discriminative score is |Accuracy-0.5|, and if the score is low, classification is difficult, so the original data and the synthesized data are decided to be similar. The predictive score measures the effectiveness of the synthesized data using the train-synthesis-and-test-real (TSTR) method. After training a model that predicts the next step using the synthesized data, the mean absolute error (MAE) is calculated between the predicted values and the ground-truth values in test data. If the MAE is small, the model trained using the synthesized data is decoded to be similar to the original data. For qualitative evaluation, the synthetic data is visualized with the original data. There are two methods for visualization. One is to project original

---

[1]A well-posed problem means i) its solution uniquely exists, and ii) its solution continuously changes as input data changes.

Table 1: Regular time series

| | Method | Sines | Stocks | Energy | MuJoCo |
|---|---|---|---|---|---|
| Discriminative Score | GT-GAN | .012±.014 | **.077±.031** | **.221±.068** | **.245±.029** |
| | TimeGAN | **.011±.008** | .102±.021 | .236±.012 | .409±.028 |
| | RCGAN | .022±.008 | .196±.027 | .336±.017 | .436±.012 |
| | C-RNN-GAN | .229±.040 | .399±.028 | .499±.001 | .412±.095 |
| | T-Forcing | .495±.001 | .226±.035 | .483±.004 | .499±.000 |
| | P-Forcing | .430±.227 | .257±.026 | .412±.006 | .500±.000 |
| | WaveNet | .158±.011 | .232±.028 | .397±.010 | .385±.025 |
| | WaveGAN | .277±.013 | .217±.022 | .363±.012 | .357±.017 |
| Predictive Score | GT-GAN | .097±.000 | .040±.000 | .312±.002 | **.055±.000** |
| | TimeGAN | **.093±.019** | .038±.001 | **.273±.004** | .082±.006 |
| | RCGAN | .097±.001 | .040±.001 | .292±.005 | .081±.003 |
| | C-RNN-GAN | .127±.004 | **.038±.000** | .483±.005 | .055±.004 |
| | T-Forcing | .150±.022 | .038±.001 | .315±.005 | .142±.014 |
| | P-Forcing | .116±.004 | .043±.001 | .303±.006 | .102±.013 |
| | WaveNet | .117±.008 | .042±.001 | .311±.005 | .333±.004 |
| | WaveGAN | .134±.013 | .041±.001 | .307±.007 | .324±.006 |
| | Original | .094±.001 | .036±.001 | .250±.003 | .031±.003 |

Table 2: Irregular time series (30% dropped)

| | Method | Sines | Stocks | Energy | MuJoCo |
|---|---|---|---|---|---|
| Discriminative Score | GT-GAN | **.363±.063** | **.251±.097** | **.333±.063** | **.249±.035** |
| | TimeGAN-△t | .494±.012 | .463±.020 | .448±.027 | .471±.016 |
| | RCGAN-△t | .499±.000 | .436±.064 | .500±.000 | .500±.000 |
| | C-RNN-GAN -△t | .500±.000 | .500±.001 | .500±.000 | .500±.000 |
| | T-Forcing-△t | .395±.063 | .305±.002 | .477±.011 | .348±.041 |
| | P-Forcing-△t | .344±.127 | .341±.035 | .500±.000 | .493±.010 |
| | TimeGAN-D | .496±.008 | .411±.040 | .479±.010 | .463±.025 |
| | RCGAN-D | .500±.000 | .500±.000 | .500±.000 | .500±.000 |
| | C-RNN-GAN-D | .500±.000 | .500±.000 | .500±.000 | .500±.000 |
| | T-Forcing-D | .408±.087 | .409±.051 | .347±.046 | .494±.004 |
| | P-Forcing-D | .500±.000 | .480±.060 | .491±.020 | .500±.000 |
| Predictive Score | GT-GAN | **.099±.004** | **.021±.003** | **.066±.001** | **.048±.001** |
| | TimeGAN-△t | .145±.025 | .087±.001 | .375±.011 | .118±.032 |
| | RCGAN-△t | .144±.028 | .181±.014 | .351±.056 | .433±.021 |
| | C-RNN-GAN-△t | .754±.000 | .091±.007 | .500±.000 | .447±.000 |
| | T-Forcing-△t | .116±.002 | .070±.013 | .251±.000 | .056±.001 |
| | P-Forcing-△t | .102±.002 | .083±.018 | .255±.001 | .089±.011 |
| | TimeGAN-D | .192±.082 | .105±.053 | .248±.024 | .098±.006 |
| | RCGAN-D | .388±.113 | .523±.020 | .409±.020 | .361±.073 |
| | C-RNN-GAN-D | .664±.001 | .345±.002 | .440±.000 | .457±.001 |
| | T-Forcing-D | .100±.002 | .027±.002 | .090±.001 | .100±.061 |
| | P-Forcing-D | .154±.004 | .079±.008 | .147±.001 | .173±.002 |
| | Original | .071±.004 | .011±.002 | .045±.001 | .041±.002 |

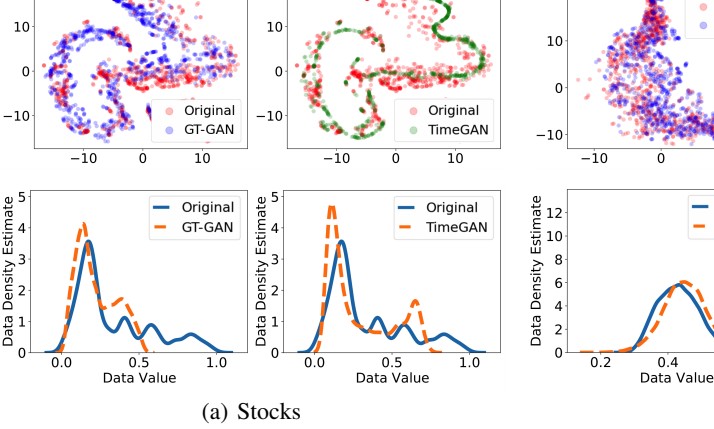
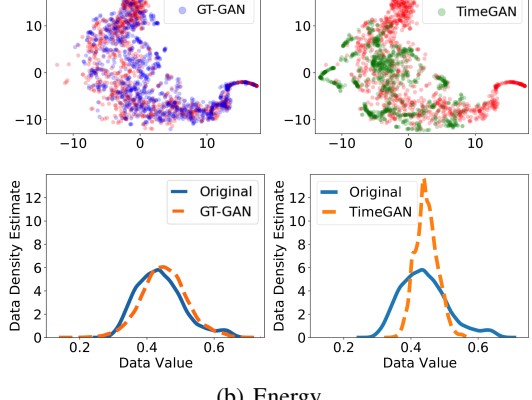

(a) Stocks          (b) Energy

Figure 4: Visualizations and distributions of the regular time series synthesized by GT-GAN and TimeGAN

and synthetic data in a two dimensional space using t-SNE [Van der Maaten and Hinton, 2008]. The other one is the kernel density estimation to draw data distributions.

## 4.2 Experimental results

**Regular time series synthesis** In Table 1, we list the results of the regular time series synthesis. GT-GAN shows better performance on most cases than TimeGAN, the previous state-of-the-art model. As shown in the 1st row in Fig. 4, GT-GAN covers original data areas better than TimeGAN. In addition, the 2nd row in Fig. 4 is the distributions of the fake data generated by GT-GAN and TimeGAN. The synthesized data's distributions from GT-GAN are more similar to those of the original data than TimeGAN, which shows the efficacy of the *explicit* likelihood training of GT-GAN against the *implicit* likelihood training of TimeGAN.

**Irregular time series synthesis** In Tables 2, 3, and 4, we list the results of the irregular time series synthesis. GT-GAN shows better discriminative and predictive scores than other baselines in all cases. In Table 2, where we drop random 30% of observations from each time series sample, GT-GAN shows the best outcomes, outperforming TimeGAN by large margins. Baselines modified with GRU-△t and those with GRU-Decay show comparable results and it is hard to say one is better than the other in this table.

Table 3: Irregular time series (50% dropped)    Table 4: Irregular time series (70% dropped)

| | Method | Sines | Stocks | Energy | MuJoCo |
|---|---|---|---|---|---|
| Discriminative Score | GT-GAN | **.372±.128** | **.265±.073** | **.317±.010** | **.270±.016** |
| | TimeGAN-$\triangle t$ | .496±.008 | .487±.019 | .479±.020 | .483±.023 |
| | RCGAN-$\triangle t$ | .406±.165 | .478±.049 | .500±.000 | .500±.000 |
| | C-RNN-GAN-$\triangle t$ | .500±.000 | .500±.000 | .500±.000 | .500±.000 |
| | T-Forcing -$\triangle t$ | .408±.137 | .308±.010 | .478±.011 | .486±.005 |
| | P-Forcing-$\triangle t$ | .428±.044 | .388±.026 | .498±.005 | .491±.012 |
| | TimeGAN-D | .500±.000 | .477±.021 | .473±.015 | .500±.000 |
| | RCGAN-D | .500±.000 | .500±.000 | .500±.000 | .500±.000 |
| | C-RNN-GAN-D | .500±.000 | .500±.000 | .500±.000 | .500±.000 |
| | T-Forcing-D | .430±.101 | .407±.034 | .376±.046 | .498±.001 |
| | P-Forcing-D | .499±.000 | .500±.000 | .500±.000 | .500±.000 |
| Predictive Score | GT-GAN | **.101±.010** | **.018±.002** | **.064±.001** | **.056±.003** |
| | TimeGAN-$\triangle t$ | .123±.040 | .058±.003 | .501±.008 | .402±.021 |
| | RCGAN-$\triangle t$ | .142±.005 | .094±.013 | .391±.014 | .277±.061 |
| | C-RNN-GAN-$\triangle t$ | .741±.026 | .089±.001 | .500±.000 | .448±.001 |
| | T-Forcing-$\triangle t$ | .379±.029 | .075±.032 | .251±.000 | .069±.002 |
| | P-Forcing-$\triangle t$ | .120±.005 | .067±.014 | .263±.003 | .189±.026 |
| | TimeGAN-D | .169±.074 | .254±.047 | .339±.029 | .375±.011 |
| | RCGAN-D | .519±.046 | .333±.044 | .250±.010 | .314±.023 |
| | C-RNN-GAN-D | .754±.000 | .273±.000 | .438±.000 | .479±.000 |
| | T-Forcing-D | .104±.001 | .038±.003 | .090±.000 | .113±.001 |
| | P-Forcing-D | .190±.002 | .089±.010 | .198±.005 | .207±.008 |
| | Original | .071±.004 | .011±.002 | .045±.001 | .041±.002 |

| | Method | Sines | Stocks | Energy | MuJoCo |
|---|---|---|---|---|---|
| Discriminative Score | GT-GAN | **.278±.022** | **.230±.053** | **.325±.047** | **.275±.023** |
| | TimeGAN-$\triangle t$ | .500±.000 | .488±.009 | .496±.008 | .494±.009 |
| | RCGAN-$\triangle t$ | .433±.142 | .381±.086 | .500±.000 | .500±.000 |
| | C-RNN-GAN-$\triangle t$ | .500±.000 | .500±.000 | .500±.000 | .500±.000 |
| | T-Forcing-$\triangle t$ | .374±.087 | .365±.027 | .468±.008 | .428±.022 |
| | P-Forcing-$\triangle t$ | .288±.047 | .317±.019 | .500±.000 | .498±.003 |
| | TimeGAN-D | .498±.006 | .485±.022 | .500±.000 | .492±.009 |
| | RCGAN-D | .500±.000 | .500±.000 | .500±.000 | .500±.000 |
| | C-RNN-GAN-D | .500±.000 | .500±.000 | .500±.000 | .500±.000 |
| | T-Forcing-D | .436±.067 | .404±.068 | .336±.032 | .493±.005 |
| | P-Forcing-D | .500±.000 | .449±.150 | .494±.011 | .499±.000 |
| Predictive Score | GT-GAN | **.088±.005** | **.020±.005** | **.076±.001** | **.051±.001** |
| | TimeGAN-$\triangle t$ | .734±.000 | .072±.000 | .496±.000 | .442±.000 |
| | RCGAN-$\triangle t$ | .218±.072 | .155±.009 | .498±.000 | .222±.041 |
| | C-RNN-GAN-$\triangle t$ | .751±.014 | .084±.002 | .500±.000 | .448±.001 |
| | T-Forcing-$\triangle t$ | .113±.001 | .070±.022 | .251±.000 | .053±.002 |
| | P-Forcing-$\triangle t$ | .123±.004 | .050±.002 | .285±.006 | .117±.034 |
| | TimeGAN-D | .752±.001 | .228±.000 | .443±.000 | .372±.089 |
| | RCGAN-D | .404±.034 | .441±.045 | .349±.027 | .420±.056 |
| | C-RNN-GAN-D | .632±.001 | .281±.019 | .436±.000 | .479±.001 |
| | T-Forcing-D | .102±.001 | .031±.002 | .091±.000 | .114±.003 |
| | P-Forcing-D | .278±.045 | .107±.009 | .193±.006 | .191±.005 |
| | Original | .071±.004 | .011±.002 | .045±.001 | .041±.002 |

(a) Stocks          (b) Energy

Figure 5: Visualizations and distributions of the irregular time series (70% dropped) by GT-GAN, T-Forcing-△t and T-Forcing-D

In Table 3 (50% dropped), many baselines do not show reasonable synthesis quality, e.g., TimeGAN-D, TimeGAN-$\triangle t$, RCGAN-D, C-RNN-GAN-D, and C-RNN-GAN-$\triangle t$ have a discriminative score of 0.5. Surprisingly, T-Forcing-D, T-Forcing-$\triangle t$, P-Forcing-D, and P-Forcing-$\triangle t$ work well in this case. However, our model clearly shows the best performance in all datasets. Baselines modified with GRU-$\triangle t$ show slightly better than them modified with GRU-Decay in this case.

Finally, Table 4 (70% dropped) shows the results of the most challenging experiments in our paper. All baselines do not work well because of the high dropping rate. T-Forcing-D, T-Forcing-$\triangle t$, P-Forcing-D, and P-Forcing-$\triangle t$, which showed reasonable performance with a dropping rate no larger than 50%, do not work

Table 5: Ablation study for training options. Refer to Appendix E for other ablation studies with irregular time series.

| | Method (Regular) | Sines | Stocks | Energy | MuJoCo |
|---|---|---|---|---|---|
| Disc. | GT-GAN | **.012** | **.077** | **.221** | **.245** |
| | w/o Eq. (8) | .023 | .159 | .356 | .278 |
| | w/o pre-training | .046 | .175 | .312 | .290 |
| Pred. | GT-GAN | .097 | .040 | .312 | .055 |
| | w/o Eq. (8) | .097 | .043 | .315 | .057 |
| | w/o pre-training | **.096** | **.038** | **.299** | **.052** |

well in this case. This shows that they are vulnerable to highly irregular time series data. Other GAN-based baselines are vulnerable as well. Our method greatly outperforms all existing methods, e.g., a discriminative score of 0.278 by GT-GAN vs. 0.436 by T-Forcing-D vs. 0.288 by P-Forcing-$\triangle t$ for Sines, and a predictive score of 0.051 by GT-GAN vs. 0.114 by T-Forcing-D vs. 0.053 by T-Forcing-$\triangle t$ for MuJoCo. Fig 5 visually compare our method and the best performing baseline for

Table 6: Ablation study for model architecture in MuJoCo.

| Energy | GT-GAN (w.o. AE) | | GT-GAN (Flow only) | | GT-GAN (AE only) | | GT-GAN (Full model) | |
|---|---|---|---|---|---|---|---|---|
| Metric | Disc. | Pred. | Disc. | Pred. | Disc. | Pred. | Disc. | Pred. |
| 30% dropped | .500 | .054 | .467 | .156 | .495 | .162 | **.249** | **.048** |
| 50% dropped | .500 | .064 | .457 | .111 | .495 | .162 | **.270** | **.056** |
| 70% dropped | .500 | .066 | .455 | .107 | .496 | .146 | **.275** | **.051** |

the dropping rate of 70% — figures for other dropping rates and data are in Appendix I and similar patterns are observed in them.

**Ablation & sensitivity analyses** GT-GAN is characterized by the MLE training with the negative log-density in Eq. (8), and the pre-training step of the encoder and decoder. Table 5 shows the results of various GT-GAN modifications with some training mechanism removed. The model using the negative log-density training shows better performance than the model not using it. That is, the MLE training makes the synthetic data more like the real data. When the pre-trained autoencoder is not used, the predictive score is better than GT-GAN. However, the discriminative score is the worst.

In Table 6, we alter the architecture for our model. We modify our proposed GT-GAN model by removing its sub-parts to create simpler ablation models: i) In the first ablation model, we remove the autoencoder and perform the adversarial training only with our generator and discriminator, denoted "GT-GAN (w.o. AE)". In other words, our generation directly outputs raw observations (instead of hidden vectors), which will be fed into

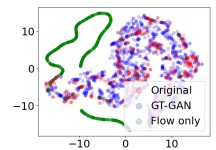 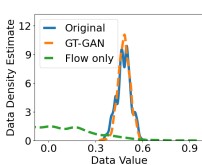

Figure 6: Visualization and distribution of MuJoCo (70% dropped) by GT-GAN and Flow only

our GRU-ODE-based discriminator. ii) The second ablation model, denoted "GT-GAN (Flow only)", has only our CTFP-based generator and we train it with the maximum likelihood training — we note that this construction is the same as training flow-based models. This model is equivalent to the original CTFP model [Deng et al., 2020]. iii) The third ablation model has only the autoencoder, denoted "GT-GAN (AE only)". However, we convert it to a variational autoencoder (VAE) model. In the full GT-GAN model, the encoder produces a set of hidden vectors $\{(t_i, \mathbf{h}_i^{real})\}_{i=0}^N$. In this ablation model, however, this is changed to $\{(t_i, \mathcal{N}(\mathbf{h}_i^{real}, \mathbf{1}))\}_{i=0}^N$, where $\mathcal{N}(\mathbf{h}_i^{real}, \mathbf{1})$ means the unit Gaussian centered at $\mathbf{h}_i^{real}$. The decoder is the same as its full model. We use the variational training for this model. Among the ablation models, GT-GAN (Flow only) outperforms the discriminator score in most cases. However, our full model is clearly the best in all cases. Our study shows that the ablation models of GT-GAN do not perform as well as its full model if any parts are missing, as shown in Fig 6. Refer to Appendix E for the ablation studies with other datasets.

The hyperparameters that significantly affect model performance are the absolute tolerance (atol), the relative tolerance (rtol), and the period of the MLE training ($P_{MLE}$) for the generator. The atol and rtol determine the error control performed by the ODE solvers in CTFPs. We test with various options of the hyperparameters in Appendix F. We found that there is an appropriate error tolerance (atol, rtol) depending on the data input size. For example, the datasets with small input sizes (i.e., Sines, Stocks) have good discriminator scores with (1e-2, 1e-3), and the datasets with large input sizes (i.e., Energy, MuJoCo) show good results with (1e-3, 1e-2).

## 5  Conclusions

Time series synthesis is an important research topic in deep learning and had been separately studied for regular or irregular time series synthesis. However, there are still no existing generative models that can handle both regular and irregular time series without model changes. Our proposed method, GT-GAN, is based on various advanced deep learning technologies, ranging from GANs to NODEs, and NCDEs, and is able to process all possible types of time series without any changes in its model architecture and parameters. Our experiments, which incorporate various synthetic and real-world datasets, prove the efficacy of the proposed method. In our ablation studies, only our full method without any missing parts shows reasonable synthesis capabilities. The limitations and societal impacts of our proposed model are in Appendix O.

## Acknowledgments and Disclosure of Funding

Noseong Park is the corresponding author. This work was partly supported by the Yonsei University Research Fund of 2022 (10%), the Institute of Information & Communications Technology Planning & Evaluation (IITP) grant funded by the Korean government (MSIT) (No. 2020-0-01361, Artificial Intelligence Graduate School Program at Yonsei University, 10%, and No. 2022-0-00113, Developing a Sustainable Collaborative Multi-modal Lifelong Learning Framework, 70%), and the LG Display research fund (C2022000673, 10%).

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
