# OpenReview forum: "GT-GAN: General Purpose Time Series Synthesis with Generative Adversarial Networks"
_NeurIPS.cc/2022/Conference — NeurIPS 2022 Accept_

### Official Review · Reviewer_Besg · 2022-07-02

**Rating:** 5
**Confidence:** 3
**Soundness:** 3 good
**Presentation:** 2 fair
**Contribution:** 2 fair

**Summary:**

This paper proposes GT-GAN, a framework for synthesizing regular and irregular time series with various techniques such as generative adversarial networks, auto-encoders, neural ordinary differential equations, neural controlled differential equations, and continuous time-flow processes. Experiments show that this method can outperform baselines in regular and irregular time series synthesis. Ablation studies show that each part of the model is critical to the synthesis performance.

**Questions:**

What are the difficulties of synthesizing regular and irregular time series in a single framework? How does the proposed method solve the difficulties? In other words, what parts of the method enable it to synthesize both regular and irregular time series? Since the author regards the ability to synthesize both regular and irregular time series as the core contribution of this work, it is better to clarify it more clearly.

In Table 5, GT-GAN without pre-training performs better than GT-GAN in terms of the predictive score, which contradicts the discriminative score. Can the authors explain this issue?

**Ethics Review Area:**

["I don’t know"]

**Limitations:**

The authors have discussed the limitations and potential negative societal impact of their work.

**Strengths And Weaknesses:**

Strengths
+ The studied problem is valuable for the field of machine learning.
+ The proposed method outperforms all baselines across regular and irregular time series synthesis qualitatively and quantitatively according to the conducted experiments.
+ The experiments are abundant, and the experimental settings are clearly stated.

Weaknesses
- The technical contributions are limited to this conference. The method proposed appears to be a combination of existing models. And some combinations of these models have been studied, e.g., Flow-GAN [Grover et al., 2017] combines maximum likelihood and GAN training.
- The proposed method lacks motivation. It is unclear to me that why we need so many technologies together to achieve the goal.
- The description of the training method is not clear to me. It would be better to give an algorithm pseudocode.

---

> ### Author Response · Authors · 2022-08-02
> **Comment**
>
> **Q1. The description of the training method is not clear to me. It would be better to give an algorithm pseudocode.**
>
> We added the training algorithm of GT-GAN in Appendix J.
>
> **Q2. The proposed method lacks motivation. It is unclear to me why we need so many technologies together to achieve the goal. And, What are the difficulties of synthesizing regular and irregular time series in a single framework? How does the proposed method solve the difficulties? In other words, what parts of the method enable it to synthesize both regular and irregular time series?**
>
> We will update our motivation if accepted. Let me try to clarify it in this comment first. Processing irregular time series is an independent research topic that is distinguished from processing regular time series. GRU-Δt, GRU-D, and NCDE are three of the most representative studies on irregular time series. In particular, recent continuous-time methods show strong points in processing irregular time series [1-7]. They are all designed especially for irregular time series.
>
> [1] Che, Z., Purushotham, S., Cho, K., Sontag, D., & Liu, Y. (2018). Recurrent neural networks for multivariate time series with missing values. Scientific reports
> [2] Rubanova, Y., Chen, R. T., & Duvenaud, D. K. (2019). Latent ordinary differential equations for irregularly-sampled time series. Advances in neural information processing systems
> [3] De Brouwer, E., Simm, J., Arany, A., & Moreau, Y. (2019). GRU-ODE-Bayes: Continuous modeling of sporadically-observed time series. Advances in neural information processing systems
> [4] Herrera, C., Krach, F., & Teichmann, J. (2020). Neural jump ordinary differential equations: Consistent continuous-time prediction and filtering. arXiv preprint arXiv:2006.04727.
> [5] Kidger, P., Morrill, J., Foster, J., & Lyons, T. (2020). Neural controlled differential equations for irregular time series. Advances in Neural Information Processing Systems
> [6] Jhin, S. Y., Lee, J., Jo, M., Kook, S., Jeon, J., Hyeong, J., ... & Park, N. (2022). EXIT: Extrapolation and Interpolation-based Neural Controlled Differential Equations for Time-series Classification and Forecasting. In Proceedings of the ACM Web Conference
> [7] Schirmer, M., Eltayeb, M., Lessmann, S., & Rudolph, M. (2022). Modeling irregular time series with continuous recurrent units. PMLR.
>
>
> To achieve our goal, i.e., a unified time series generative framework for regular and irregular time series, we extensively use continuous methods when designing our autoencoder and GAN parts. We already conducted ablation studies on how each part contributes to the final model performance, which shows the efficacy of our specific model design. When they are integrated into a single framework, the reported performance can be obtained. We also report some new results about this in the comment for the reviewer xbxz. You can also refer to them.
>
> **Q3. In Table 5, GT-GAN without pre-training performs better than GT-GAN in terms of the predictive score, which contradicts the discriminative score. Can the authors explain this issue?**
>
> In Appendix K, we visualize GT-GAN and GT-GAN(w/o pre-training). As shown, GT-GAN has a much larger sampling diversity than GT-GAN(w/o pre-training). In general, GT-GAN(w/o pre-training) produces fake data in a narrow region following the original data distribution, i.e., a kind of mode-collapse. We conjecture that this helps improve the predictive score although its discrimination score is mediocre. Recall that following the predictive task evaluation protocol of TimeGAN, we train another RNN-based forecasting model with fake data for this predictive task. The base forecasting model may better learn the data in the narrow region by GT-GAN(w/o pre-training). Similar patterns are observed in Fig. 5 for GT-GAN and TimeGAN.
>
> **Q4. Comparison with Flow-GAN [Grover et al., 2017]**
>
> FlowGAN is influential work, and many papers follow its direction. FlowGAN combines a flow-based generator with a discriminator and performs log-density training and adversarial training. However, FlowGAN does not use an autoencoder. In order to achieve our goal, i.e., building general purpose time series synthesis models, our autoencoder plays a vital role, i.e., our encoder and decoder are able to process both regular and irregular time series. Please understand that we successfully designed a general-purpose time series synthesis method, being inspired by FlowGAN.

---

> > ### Comment · Reviewer_Besg · 2022-08-08
> > **Response to Authors**
> >
> > I would like to thank the authors for their response.
> >
> > I am not fully convinced by the author's explanation of Q3. I do not think that a forecasting model trained on samples with low diversity (GTGAN w/o pre-training) would yield a better predictive score than a forecasting model trained on samples with close diversity to the original data distribution (GTGAN). In other words, the closer the distributions of the training set and the test set are, the better the predictive score of the forecasting model should be.
> >
> > Also, I still have concerns about the motivation of the method. In my opinion, regular time series is a special case of irregular time series. Therefore, irregular time series synthesis methods are able to synthesize regular time series as well. I acknowledge that the proposed method achieves superior results over competing methods according to the experiments, but the advantages come at the cost of a more complex model design and training algorithm, which is a combination of existing technologies. I would like the authors to explain in detail what specific issue each part addresses to improve the performance, instead of showing the results in the ablation experiments without analysis or explanations.

---

> > > ### Author Response · Authors · 2022-08-08
> > > **Response**
> > >
> > > We greatly appreciate your follow-up questions.
> > >
> > > **Discriminative vs. Predictive Scores**
> > >
> > > If you carefully look at those t-SNE figures (especially Figs. 5 and 21), GT-GAN generates samples a little out of the original data distribution whereas GT-GAN w/o pre-training has severe mode-collapse problems (i.e., generating in a narrow region). For now, we conjecture that those samples a little outside the original data distribution make the prediction tasks' scores a little low. However, note that GT-GAN can successfully recall almost the entire data region.
> > >
> > > **Irregular vs. Regular**
> > >
> > > Thanks for pointing this out. The role of each part is as follows:
> > >
> > > 1. The neural CDE-based encoder is able to encode a regular/irregular time series sample into a regular/irregular hidden vector sequence. Neural CDEs are sometimes called *continuous* RNNs and are specially designed for the representation learning of irregular time series. As reported in our first email, generation quality is severely degraded when this network is substituted with GRU-Δt.
> > >
> > > 2. The GRU-ODE-based decoder is able to decode a regular/irregular hidden vector sequence into a regular/irregular time series sample. One beauty of this decoder is, as shown in Fig. 2 in our main paper, that the sampling time point and the sample length can be freely determined by users.
> > >
> > > 3. The CTFP-based invertible generator was intentionally selected by us since we can perform both the log-likelihood and the adversarial training together. Since this network is a key part of our model, we wanted to use the two different training paradigms. Our ablation studies about the log-likelihood and supervised-learning training justify our design selection.
> > >
> > > 4. The GRU-ODE-based discriminator is able to process regular/irregular time series. Unlike the encoding task of the neural CDE-based encoder, we observed faster and better results with the GRU-ODE-based discriminator. Moreover, neural CDEs require interpolation of input as a pre-processing. We can do this for real data before training. However, it is hard to perform dynamically for the fake hidden vector sequence due to its excessive computation amount. In particular, it significantly delays the overall training process if we use a neural CDE-based discriminator.
> > >
> > > In general, our key design points lie in utilizing i) the continuous-time method-based autoencoder, and ii) the CTFP-based generator. Therefore, we can stabilize the generating performance for complicated irregular time series as well. We also note that this area, i.e., irregular time series generation, is under-explored, and therefore, we wanted to contribute to our research community by designing a general-purpose time series synthesis method.
> > > Thanks for reading our comments and providing us with guidance.

---

> > > > ### Comment · Reviewer_Besg · 2022-08-08
> > > > **Response to Authors**
> > > >
> > > > I would like to thank the authors for their quick response. I decided to raise my score to 5 (Borderline Accept), considering the potential impact of the proposed method in the field of time series synthesis. Nevertheless, I reserve my concerns about Q3 because the authors did not give rigorous proof but only conjecture. I suggest that the authors elaborate on the motivation and role for designing each part of the proposed method in the updated version. For example, it would be better to intuitively answer why performing both the log-likelihood and the adversarial training together can achieve better results in time series synthesis tasks.

---

> > > > > ### Author Response · Authors · 2022-08-08
> > > > > **Thanks for consideration.**
> > > > >
> > > > > Dear Reviewer Besg,
> > > > >
> > > > > Thanks for your consideration. Following your comments, we will strengthen our motivations and the role of each part. We will upload the revised manuscript as soon as possible. We are sorry that we just deliver a conjecture about Q3 but will keep inspecting Q3.
> > > > >
> > > > > Best regards,
> > > > >
> > > > > Authors.

---

> ### Author Response · Authors · 2022-08-08
> **We need your comments.**
>
> Dear Reviewer Besg,
>
> Could you please leave follow-up questions if any? We can answer them.
>
> Best regards,
> Authors

---

### Official Review · Reviewer_xbxz · 2022-07-14

**Rating:** 7
**Confidence:** 2
**Soundness:** 2 fair
**Presentation:** 2 fair
**Contribution:** 3 good

**Summary:**

The paper presents a time series GAN model that uses neural ODEs and continues time-flow processes to generate synthetic time series from regular and irregular data (irregular means time series data having missing observations). Specifically, the authors use neural ordinary differential equations, neural controlled differential equations, and continuous time-flow processes. The authors evaluate their approach against standard baselines and datasets and conduct an ablation study of different components. In particular, their method outperforms baselines in the irregular data setting with missing observations.

**Questions:**

1. ODEs (and their variants) are a natural choice for modeling data with missing observations and this paper provides further evidence for this. However, the overall architecture is very similar to TimeGAN and therefore it is hard to judge how much gain is coming from the architecture itself or by replacing the standard RNN units with continuous counterparts. In particular, I one ablation that would be interesting is to replace the continuous units with GRU-$\Delta$-t or GRU-D. How much performance does the model loose when using these units in GT-GAN? Likewise, how would the baselines perform (in particular TimeGAN) if the corresponding ODE units are used instead of the GRU-$\Delta$-t or GRU-D? This would help to judge how much performance actually stems from the use of continuous units.

2. The overall architecture is very similar to TimeGAN in the sense that it uses an autoencoder path, a GAN path and a way to link the two (in this case, the inversion/recovery and log density of the hidden representation from the autoencoder). However, this component is not analyzed in the ablation study. How does the model perform when removing this log-density path? Could this path be replaced with a supervised loss between the hidden (fake) representation of the CTFP and the autoencoder? Since this component is the biggest differentiator between the proposed architecture and TimeGAN (on top of the continuous units, of course) it would be interesting to measure this effect.

3. The authors do not provide details on how the hyperparameters of GT-GAN are tuned. I would kindly ask the authors to discuss their tuning method in the paper.

**Limitations:**

I appreciate that the authors include a limitations statement and discusses that the model contains many hyperparameters and is difficult to train because of that. While this might be the case it is a fairly general statement. It would be more interesting for the reader to understand what specific limitations arise from using continuous units (if there are any). Also, the authors discuss the training time in the Appendix and I think this could be moved to the main text.

**Strengths And Weaknesses:**

The main contribution in this paper is the use of ODE/continuous time-flow processes into a GAN framework for time series synthesis. ODEs and CTFPs are a natural choice when learning from data with missing observations. The overall architecture is close to TimeGAN and uses an autoencoder path (with reconstruction loss) and an adversarial path with a discriminator that is trained on a standard adversarial loss (the individual components are continuous units, though). An interesting addition is the use of the invertible CTFP to use the hidden representation from the autoencoder to train the "log-density" path by converting the hidden representation into noise vectors and then again into a hidden representation by the CTFP and computing its log probability.

The overall architecture of the model is reasonable and the evaluation is sound overall. However, since the main contribution of this paper is using continuous layers for irregular time series synthesis, I would like to propose additional experiments to strengthen the paper.

---

> ### Author Response · Authors · 2022-08-02
> **Comment**
>
> **Q1. ODEs (and their variants) are a natural choice for modeling data with missing observations and this paper provides further evidence for this. However, the overall architecture is very similar to TimeGAN and therefore it is hard to judge how much gain is coming from the architecture itself or by replacing the standard RNN units with continuous counterparts. In particular, I one ablation that would be interesting is to replace the continuous units with GRU-Δ-t or GRU-D. How much performance does the model loose when using these units in GT-GAN? Likewise, how would the baselines perform (in particular TimeGAN) if the corresponding ODE units are used instead of the GRU-Δ-t or GRU-D? This would help to judge how much performance actually stems from the use of continuous units.**
>
> We executed two experiments based on your advice. First, we experimented by replacing the encoder of TimeGAN with Neural CDE instead of GRU-Δt. In the second experiment, the encoder of GT-GAN was changed to GRU-Δt instead of Neural CDE. The results are as follows.
>
> |      Regular             | Discriminative Score | Predictive Score |
> |:------------------------:|:--------------------:|:----------------:|
> |          GT-GAN          |       .077         |   .040         |
> | TimeGAN (CDE) |  .183             |     .036       |
> | GT-GAN (GRU-Δt) |   .184            |     .041       |
>
>
> |       Irregular (30%)    | Discriminative Score | Predictive Score |
> |:------------------------:|:--------------------:|:----------------:|
> |          GT-GAN          |         .251        |       .021      |
> | TimeGAN (CDE) |      .430         |     .036       |
> | GT-GAN (GRU-Δt) |      .345         |   .022         |
>
> In the first experiment with regular data, it can be determined that the structure of our model is important, and in the second experiment with irregular data, replacing standard RNN units with continuous counterparts decreases the overall performance. Overall, our specific design of GT-GAN is sophisticatedly selected.
>
>
> **Q2.The overall architecture is very similar to TimeGAN in the sense that it uses an autoencoder path, a GAN path and a way to link the two (in this case, the inversion/recovery and log density of the hidden representation from the autoencoder). However, this component is not analyzed in the ablation study. How does the model perform when removing this log-density path? Could this path be replaced with a supervised loss between the hidden (fake) representation of the CTFP and the autoencoder? Since this component is the biggest differentiator between the proposed architecture and TimeGAN (on top of the continuous units, of course) it would be interesting to measure this effect.**
>
> In order to see the efficacy of the log-density training, we already conducted 'w/o Eq. (8)' in Table 5 in our initial submission, in which the generator is trained only with the adversarial loss. In addition to it, based on your advice, we experimented after replacing the log-density loss with a supervised loss. To obtain the supervised loss, like TimeGAN, we added a supervisor network between the encoder and decoder. The result is as follows.
>
> |      Regular             | Discriminative Score | Predictive Score |
> |:------------------------:|:--------------------:|:----------------:|
> |          GT-GAN          |       .077         |   .040         |
> | GT-GAN (Supervised Loss) |      .124         |   .037         |
>
> According to the above results, it was confirmed that even if TimeGAN's supervised loss is used, no better results than those of our original design are obtained (the predictive score is slightly improved though). In other words, this experiment confirms the importance of the log-density path in our model.
>
> **Q3. The authors do not provide details on how the hyperparameters of GT-GAN are tuned. I would kindly ask the authors to discuss their tuning method in the paper.**
>
> The absolute tolerance (atol), relative tolerance (rtol), and the period of the MLE training (PMLE) for the generator are the hyperparameters that have the biggest effect on the model performance. We executed experiments through the grid search using the hyperparameters shown in the figures in Appendix F.

---

> > ### Comment · Reviewer_xbxz · 2022-08-09
> > **Response to Authors**
> >
> > I would like to thank the authors for the additional experiments. I have raised my score to seven because these experiments are important controls. It is interesting that the variants GT-GAN (GRU-Δt) and GT-GAN (Supervised Loss) have only small impact on predictive scores while having significant impact on discriminative scores. Do the authors have any insight on why this is the case? I would appreciate a discussion on this in the final version of the paper.

---

> ### Author Response · Authors · 2022-08-08
> **Please leave questions if any.**
>
> Dear Reviewer xbxz,
>
> Please leave more questions if any. We can answer unclear points. In particular, you marked relatively lower confidence (in comparison with other reviewers) so we are ready to answer all your potential unclear points to help your understanding.
>
> Best regards,
>
> Authors

---

### Official Review · Reviewer_BsWG · 2022-07-14

**Rating:** 7
**Confidence:** 3
**Soundness:** 4 excellent
**Presentation:** 4 excellent
**Contribution:** 3 good

**Summary:**

The authors present a GAN-based architecture, called GT-GAN, for generating synthetic samples for regular and irregular time series. It comprises of two main components: (1) Autoencoder(AE) - encoder (Neural CDE) encodes time series into a latent space and decoder (GRU-ODE) recovers a continuous path from which synthetic time series is sampled. It is pre-trained using standard reconstruction loss. Decoder is trained additionally with GAN's discriminator loss. (2) Generator(CTFP) and Discriminator(GRU-ODE) networks which are trained using adversarial loss and maximum likelihood training with the log-density. Generator synthesize a time series latent vectors which is used by AE decoder to generate the time-series. This fake time series is then passed to discriminator along with true samples. The authors evaluate their method on 2 simulated and 2 real-world datasets and compare it against 7 related methods. While GT-GAN doesn’t always have a better performance for regular time series, however, it outperforms all other methods for irregular time series. The authors also present an ablation study to understand the role of each loss term and show that each loss component is required for the superior performance of the model.

**Questions:**

Good work. I’m just asking this out of curiosity. Since GT-GAN learns the temporal dynamics of irregular time series well, do you think if the model can be modified to do forecasting (which is history conditioned time series generation) for irregular time series?

**Limitations:**

As the authors mentioned in the limitations, the proposed method has many hyperparameters and it often becomes difficult to tune with standard methods like Grid Search. One way to deal with it is tying AutoML libraries like SyneTune which provides several state-of-the-art distributed hyperparameter optimizers (HPO).

**Strengths And Weaknesses:**

1. First of all, the paper is very well-written and provide good amount of details to understand the motivation of the work, and the proposed method.
2. The papers presents a through comparison with several related methods. The authors made extra efforts to modify competing methods for making compatible with irregular time series, and therefore, ensuring a fair comparison.
3. The evaluation shows the superiority of the method both quantitatively and qualitatively, especially for irregular time series.

---

> ### Author Response · Authors · 2022-08-02
> **Comment**
>
> **Q1. Good work. I’m just asking this out of curiosity. Since GT-GAN learns the temporal dynamics of irregular time series well, do you think if the model can be modified to do forecasting (which is history-conditioned time series generation) for irregular time series?**
>
> Forecasting is similar to conditional generation, i.e., generating future observations conditioned on past observations. For this, we need to modify our model architecture. Therefore, we also think that it is possible after changing our generator to a conditional generator.

---

### Author Response · Authors · 2022-08-09
**Notification of uploading a revised paper**

Dear All Reviewers,

First of all, thanks for your comments. We could significantly improve our paper during the discussion period. We revised the following points and uploaded a new version:

1. We revised Introduction to strengthen our motivations for the general purpose time series synthesis model.

2. We added a training algorithm in Appendix J.

3. We also added additional experimental results from Appendix K.

4. We describe the role of each network in detail in Appendix M.

Best regards,

Authors

---

### Meta-Review · Area_Chair_fGxd · 2022-08-26

**Recommendation:** Accept
**Confidence:** Certain

**Metareview:**

This paper presents a new model for time series data that can handle data sampled at irregular time intervals.  The proposed model makes extensive use of continuous time processes in a GAN framework.  Experiment results show that the proposed model consistently outperforms existing approaches.  All reviewers lean on the accept side and I support that consensus.

**Award:**

No

---

### Decision · Program_Chairs · 2022-09-14

Accept